# Impact of Open Dialogue about Complementary Alternative Medicine—A Phase II Randomized Controlled Trial

**DOI:** 10.3390/cancers14040952

**Published:** 2022-02-14

**Authors:** Mette Stie, Charlotte Delmar, Birgitte Nørgaard, Lars Henrik Jensen

**Affiliations:** 1Department of Oncology, University Hospital of Southern Denmark, 7100 Vejle, Denmark; lars.henrik.jensen@rsyd.dk; 2Department of Regional Health Research, Faculty of Health Sciences, University of Southern Denmark, 5000 Odense, Denmark; 3Department of Public Health, Research Unit for Nursing and Health Care, Aarhus University, 8000 Aarhus, Denmark; cd@ph.au.dk; 4Department of Public Health, University of Southern Denmark, 5000 Odense, Denmark; binorgaard@health.sdu.dk

**Keywords:** complementary alternative medicine, communication, integrative, oncology, patient safety, quality of life

## Abstract

**Simple Summary:**

A large number of patients with cancer use complementary alternative medicine (CAM), such as diet supplements, massage and acupuncture, as an adjunct to conventional cancer treatment and care. Some types of CAM reduce nausea and vomiting, pain, fear, fatigue and depression, but CAM may also cause new symptoms and side effects. Therefore, it is crucial that cancer patients receive professional guidance on how to use CAM in a safe and healthy manner. Open dialogue about CAM between patients and health professionals is, however, not an integrated part of cancer treatment and care. Therefore, the aim of our study was to assess how open dialogue, including guidance about CAM, affected patients’ safety and health when it was an integrated part of the cancer treatment and care. We found that open dialogue about CAM does not compromise patient safety and that it may improve patients’ quality of life, self-care and survival.

**Abstract:**

Complementary alternative medicine (CAM) may reduce the symptom burden of side effects to antineoplastic treatment but also cause new side effects and non-adherence to conventional treatment. The aim of this RCT was to investigate the impact of open dialogue about complementary alternative medicine (OD-CAM) on cancer patients’ safety, health and quality of life (QoL). Patients undergoing antineoplastic treatment were randomly assigned to standard care (SC) plus OD-CAM or SC alone. The primary endpoint was frequency of grade 3–4 adverse events (AE) eight weeks after enrollment. Secondary endpoints were frequency of grade 1–4 AE, QoL, psychological distress, perceived information, attitude towards and use of CAM 12 and 24 weeks after enrollment. Survival was analyzed post hoc. Fifty-seven patients were randomized to the OD-CAM group and fifty-five to the SC group. No significant difference in frequency of grade 3–4 AEs was shown. The same applied to grade 1–4 AEs and QoL, psychological distress and perceived information. A tendency towards better QoL, improved survival and a lower level of anxiety was found in the OD-CAM group. OD-CAM is not superior to SC in reducing the frequency of AEs in patients undergoing antineoplastic treatment. OD-CAM does not compromise patient safety; it may reduce psychological stress and improve QoL and overall survival.

## 1. Introduction

Worldwide, an increasing number of cancer patients use complementary alternative medicine (CAM) as an adjunct to conventional treatment and care [1,2,3]. A systematic review reported CAM use among 75% of breast cancer patients undergoing chemotherapy [4]. Among patients undergoing treatment for colorectal cancer, a Danish study has shown that 49.9% use CAM [5]. Studies including patients with various types of cancer have found a prevalence of CAM use of 39.1% [6] and 60.3% [2].

There is no evidence that CAM itself has the potential to cure or affect the cancer disease, but some studies suggest that CAM as an adjunct to conventional treatment is associated with higher survival rates [7,8,9] and that specific types of CAM are relevant as supportive therapies in managing cancer-related symptoms and side effects. Acupressure and acupuncture reduce nausea and pain [10], aromatherapy alleviates sleep and anxiety disorders [11] and massage, yoga, mindfulness and meditation have been shown to increase quality of life (QoL) and reduce stress and fatigue [12]. CAM has also been shown effective in relieving fear, fatigue and depression [13] and enhancing hope [2], self-care, self-control and empowerment [14,15]. However, the level of evidence ranges from high to low, and some CAMs include a potential risk of interaction with conventional medicine [16,17,18]. Therefore, to ensure patient safety and high-quality care, the Society for Integrative Oncology (SIO) and American Society of Clinical Oncology (ASCO) have developed clinical practice guidelines on how to practice integrative oncology (IO) [19]. IO is a patient-centered, evidence-informed field of cancer care that utilizes mind and body practices, natural products and/or lifestyle modifications from different traditions alongside conventional cancer treatments. It aims to optimize health, quality of life and clinical outcomes across the cancer continuum and to empower people to prevent cancer and become active participants before, during and beyond cancer treatment [20]. An increasing number of cancer centers in North America [21,22,23], Germany [24,25] and Italy [26] practice IO.

In Scandinavia and Denmark, IO is not integrated in daily conventional oncology care. Nevertheless, considering the high rate of CAM use among Danish cancer patients [5], there is a need for health professionals and patients to have an open dialogue about the potential benefits and harms of CAM when combined with conventional oncology treatment [27,28]. Studies have shown that open dialogue about CAM increases patient engagement, patient-centered communication and higher clinician [29] and patient satisfaction [30]. It addresses patient stress and uncertainty, reduces exposure to misleading information and enhances the patient–physician relationship, which is paramount in delivering high-quality care [31]. Still, it remains unclear whether an open dialogue about CAM (OD-CAM) actually affects patients’ safety, health and QoL. The primary aim of this phase II randomized controlled trial was therefore to investigate whether OD-CAM is superior to standard care (SC) in reducing the frequency of adverse events in patients undergoing oncology treatment. Moreover, we hypothesized that patients participating in OD-CAM would report improved QoL, reduced anxiety and depression and a higher level of perceived information compared to patients receiving SC alone. The study was designed to provide knowledge on how to conduct a safe, high-quality OD-CAM in oncology care.

## 2. Materials and Methods

### 2.1. Trial Design

This phase II, parallel group, randomized controlled trial compared the effectiveness of OD-CAM with SC in reducing adverse events (AEs) in patients undergoing oncology treatment and care. The study was prospectively registered with ClinicalTrials.gov (NCT03857776) and approved by the Danish Data Protection Agency through the Region of Southern Denmark (19-4309). According to the Committee on Health Research Ethics, their approval of the study was not required (15/42744). The procedures used in this study adhere to the tenets of the Declaration of Helsinki. The protocol was not amended during the study, and since it did not involve any risks to the patients, no interim analysis was performed. The study is reported according to the CONSORT guidelines [32].

### 2.2. Setting

The study was conducted at the Oncology Outpatient Clinic, Vejle Hospital, University Hospital of Southern Denmark, between April 2019 and July 2020. The Department of Oncology offers conventional treatment and care to adult patients with breast, gynecological, prostate, pulmonary, colorectal, anal and pancreatic cancer. Currently, no CAM treatments are offered. There are around 57,000 outpatient visits to the department each year, with 23,000 radiotherapy fractions and 9300 chemotherapy and immunotherapy treatments administered.

### 2.3. Participants

The inclusion criteria were: ≥18 years of age, diagnosis of primary cancer or recurrence within the last three months, planned antineoplastic treatment (chemotherapy, immunotherapy and/or antibody therapy), realistic plan of at least two months of treatment and life expectancy of six months. The ability to read and speak Danish was required. The exclusion criterion was participation in other trials interfering with the intervention or data collection. Eligible participants were informed and invited to participate in the study by health professionals prior to the first cycle of treatment in the outpatient clinic. Randomization into the study was based on written and orally informed consent.

### 2.4. Intervention Group (OD-CAM)

In addition to SC, patients in the intervention group participated in one or two sessions of OD-CAM facilitated by a nurse specialist who has completed the program in Fellowship in Integrative Medicine at the University of Arizona. This is a training program for health professionals in empowering individuals and communities to optimize health and well-being through evidence-based, sustainable and integrative approaches [33]. Thus, inspired by the principles of integrative medicine the nurse specialist paid careful attention to the patients’ experiences, values, beliefs, concerns and needs and provided evidence-based sources of information about which CAM treatments should be avoided or recommended. She did not provide CAM treatments. A primary caregiver participated if preferred by the patient. The number of OD-CAM sessions depended on the needs of the individual patient. The guideline for OD-CAM presented in Table 1 was inspired by the Andrew Weil Center for Integrative Medicine, University of Arizona and Schofield et al.’s recommendations [33,34].

### 2.5. Control Group (SC)

SC was defined as oncology treatment and care, including antineoplastic drugs at the outpatient clinic and, if necessary, hospitalization at the oncology inpatient clinic. SC also involved continuous assessment of the patients’ performance status, side effects, and symptoms, which were managed by specialist doctors and nurses. SC did not include any specialized consultations and guidance about CAM. Patients randomized to the control group received SC and received a pamphlet from the project nurse referring to www.kabcancer.dk (accessed on 2 January 2022). The website was developed by a team of researchers and based on systematic reviews. The website presents information about potential effects and outcomes of specific CAM treatments, such as acupuncture, antioxidant supplements, mindfulness, herbs, massage, etc. The website is accessible to the public.

### 2.6. Randomization

Based on written, orally informed consent and subsequent baseline assessment, the patients were randomly assigned to either SC plus OD-CAM or SC alone. Randomization 1:1 was computerized using Research Electronic Data Capture (REDCap) [35]. The study nurse informed the patients about the allocation. Patients randomized to OD-CAM received a letter providing the date, time and place for the OD-CAM session and guidance for preparation. Patients randomized to SC received a pamphlet referring to the website www.kabcancer.dk (accessed on 2 January 2022).

Due to the nature of the intervention, neither patients nor staff were blinded to the allocation, but patients were strongly encouraged not to disclose the allocation status at the follow-up registration of adverse events.

### 2.7. Outcome Measures and Data Collection

The primary outcome measure was the frequency of grade 3–4 AEs eight weeks after enrollment. The frequency of grade 0–4 AEs 12 and 24 weeks after enrollment were secondary outcome measures. At each follow-up (8, 12 and 24 weeks), patients’ AEs were registered by a specialist nurse according to the Common Terminology Criteria for Adverse Events (CTCAE v5) [36]. We have registered the severity of 15 common AEs, i.e., dry mouth, oral mucositis, vomiting, nausea, constipation, diarrhea, pain, peripheral motor neuropathy, peripheral sensory neuropathy, fatigue, fever, febrile neutropenia, infections, hospitalization and general discomfort. The secondary outcome measures included patient-reported QoL, level of depression and anxiety and perception of received information 12 and 24 weeks after enrollment.

To assess QoL, the validated European Organization for Research and Treatment of Cancer Quality of Life Questionnaire (EORTC QLQ C30) [37] was applied. It includes five functional scales, nine symptom scales and two global QoL scales. Patients’ perception of information received was assessed using EORTC QLQ-INFO25 [38]. It consists of 26 items organized in four hypothesized scales: information about the disease, medical tests, treatment and other services and eight single items. The level of depression and anxiety was assessed using the Hospital Anxiety and Depression Scale (HADS), which is a self-assessment scale composed of 14 items on two subscales assessing anxiety and depression symptoms in the past week [39]. The patients’ use of and attitude towards CAM was measured at baseline and 24 weeks after enrollment. All questionnaires were administered electronically.

### 2.8. Statistical Methods

The study is a randomized phase II screening trial [40] with a risk of type 1 error at 0.10 and a power of 0.80. Based on the study of Frenkel, which has shown that consultations about CAM reduces intense distress (physical problems, general well-being, chemotherapy side effects) to less than half [21], it was hypothesized that 25% of the patients in the OD-CAM group would have grade 3–4 AEs eight weeks after enrollment compared to 50% in the SC group. Under these circumstances, 92 patients were required. To account for dropouts, the total number of patients to be enrolled was 106.

The two groups were compared in all primary analyses. Demographic data are presented as counts (*n*) and proportions (%), respectively, with means and standard deviations (SD). A chi-squared test and Fisher’s exact test was applied to detect differences between the two groups in relation to AEs. The EORTC QLQ C30 and INFO25 scores were reported as means with confidence intervals and HADS scores as medians with 25th–75th percentiles. Comparison of the two groups relied on the Student’s *t*-test or Mann–Whitney’s U test.

*p*-values are reported to two decimal places. For the primary endpoint, two-sided *p*-values were used with a 0.10 level of significance. A professional academic statistician blinded to the study group assignment conducted all analyses.

## 3. Results

Of 454 patients screened for eligibility, 256 were invited to participate in the study. The remaining 198 were not invited due to restricted time resources. A total of 144 declined participation due to lack of interest (*n* = 53), lack of personal resources (*n* = 42), too many extra visits (*n* = 34), other reasons (*n* = 8) and administrative failure (*n* = 7). There were no significant differences in terms of age, sex and cancer diagnosis between the decliners and those randomized (data not shown). In total, 112 patients were randomly assigned to OD-CAM (*n* = 57) and SC (*n* = 55) (Figure 1). The groups were comparable in terms of baseline demographics, clinical characteristics and use of and attitude towards CAM (Table 2).

The reply rate in the study was high, with 87% and 98% at first follow-up (8 weeks), 94% and 86% at second follow-up (12 weeks) and 88% and 89% at third follow-up (24 weeks) for OD-CAM and SC, respectively (see Figure 1).

Patients in the intervention group participated in 0–4 OD-CAM sessions with the nurse specialist; four did not show up to the session due to lack of energy, 49 participated in one session, two had two sessions and one patient had four sessions. For one patient, data on the number of sessions was lost.

At baseline, the two groups were comparable in terms of frequency of adverse events, QoL, depression and anxiety and perceived information. During follow-up (8, 12 and 24 weeks), frequency of adverse events, level of QoL, depression and anxiety and perceived information was in general similar in the two groups.

### 3.1. Adverse Events

Eight weeks after enrollment, no significant difference between the OD-CAM and SC group was found for any type of grade 3–4 AE. The same applied to the follow-up after 12 and 24 weeks. Pooling of data at the patient level did not change this. Regarding grade 1–2 AEs, the only statistically significant differences in the follow-up period were at eight weeks with nausea (28.9 vs. 52.1%, *p* = 0.02), mouth dryness (60.0% vs. 41.7%, *p* = 0.03) and sensory neuropathy (44.4% vs. 27.7%, *p* = 0.05) in the OD-CAM and SC group, respectively (see Table 3).

### 3.2. Quality of Life

There was no significant difference in QoL between the OD-CAM and SC group at 12- and 24-week follow-ups (Figure 2). Similarly, with only a few exceptions, no significant differences were found within groups during follow-up. From baseline to 12 weeks, a significant difference was shown in the SC group in one function and two symptoms. Social functioning declined by 9.65 points (95% CI: −18.59 to −0.71), diarrhea increased by 10.53 points (95% CI: 2.31–18.74) and nausea and vomiting increased by 6.41 points (95% CI: 0.96–11.86). Within the OD-CAM group, fatigue increased by 6.80 points (95% CI: 0.36–13.24). At 24 weeks, a significant difference was only shown within the OD-CAM group. Physical functioning improved by 5.24 points (95% CI: 0.03–10.44), role functioning decreased by 9.13 (95% CI: −17.47 to −0.78) and diarrhea by 10.32 points (95% CI: −17.86 to −2.77). A remarkable increase of 4.63 points (95% CI: −0.68–9.94) in emotional functioning was shown at 24 weeks in the OD-CAM group. No statistically significant differences were found within the SC group from baseline to third follow-up (Figure 2).

### 3.3. Depression and Anxiety

As shown in Figure 3, there was no statistical difference in HADS median scores between the two groups during follow-up. A slight difference in median anxiety score was observed at the second follow-up with 3 (25th–75th percentile: 2–8) and 6.5 (25th–75th percentile: 3–9) in the OD-CAM and SC group, respectively. With respect to the depression score, the SC group reported a median score of 3 (25th–75th percentile: 1–7) and the OD-CAM group reported a median score of 2 (25th–50th percentile: 1–4) at the second follow-up.

### 3.4. Level of Perceived Information

In general, the INFO25 median scores were lower in the SC group compared to the OD-CAM group during follow-up, indicating that OD-CAM might be superior to SC in providing sufficient information. However, the differences were not statistically significant (see Figure 4).

### 3.5. Attitude towards and Use of CAM

No significant differences were found in patients’ attitudes towards CAM, either at baseline or at third follow-up (24 weeks). At baseline, 22 (40.7%) and 20 (37.0%) patients in the SC group reported a positive or very positive attitude towards CAM, respectively. At third follow-up this had changed to 12 (31.6%) and 16 (42.1%), respectively. In the OD-CAM group 33 (58.9%) and 11 (19.6%) patients reported a positive or a very positive attitude towards CAM, respectively, which changed to 23 (53.5%) and 9 (20.9%) at third follow-up.

The use of CAM in the SC group changed from 40 (74.1%) patients at baseline to 27 (62.8%) at third follow-up (24 weeks). Conversely, in the OD-CAM group, the use of CAM slightly increased from 35 (62.5%) at baseline to 27 (62.8%) at third follow-up. The differences were not statistically significant.

### 3.6. Explorative Outcomes

#### Overall Survival

Survival tended to be higher in the OD-CAM group compared to the SC group with an area under the curve (AUC) of 0.902 (95% CI: 0.85–0.95) and 0.837 (95% CI: 0.77–0.91), respectively. Thus, patients in the OD-CAM lived, on average, 0.064 (95% CI: −0.022–015) years longer than did the patients in the SC group, *p* = 0.14 (Figure 5).

## 4. Discussion

In this study, 53 out of 57 patients allocated to the OD-CAM group completed the intervention and we consider it feasible to integrate OD-CAM in standard oncology care. The high completion rate could be an expression of patients’ substantial need for reliable information and counselling about CAM as an integrated part of oncology care [21,41,42].

We found no significant difference in grade 3–4 AEs between the OD-CAM and SC group after 8, 12 and 24 weeks of follow-up. Hence, OD-CAM did not prove superior to SC in reducing the frequency of grade 3–4 AEs. We found a statistically significant difference at eight weeks in three grade 1–2 AEs. The frequency of nausea was lower in the OD-CAM group and mouth dryness and sensory neuropathy was lower in the SC group. We found a statistically significant increase of nausea and vomiting in the SC group on the EORT QLQ C30 scale, although this was at the 12-week follow-up. In addition, diarrhea increased significantly in the SC group at 12 weeks and decreased significantly in the OD-CAM group at 24 weeks on the EORTC QLQ C30 scale. However, no significant difference in diarrhea was found on the CTCAE v5 scale. Although not consistent, but supportive of similar studies [38], these findings indicate that OD-CAM has potential in alleviating nausea, vomiting and diarrhea, which is clinically relevant, since these frequent AEs are associated with great concern in patients undergoing antineoplastic treatment [43,44].

Overall, OD-CAM does not compromise patient safety, which is an important finding. In-depth interviews have shown that health professionals are reluctant to discuss CAM due to skepticism as to its efficacy and safety [45]. Learning that OD-CAM does not compromise safety may render health professionals more comfortable in discussing the issue with the patients.

We found a tendency towards better QoL in the OD-CAM group, especially at the 24-week follow-up, which is in line with other studies [46]. A possible explanation of this delayed effect may be that the first phase of the antineoplastic treatment is the most burdensome. Another reason might lie in the fact that OD-CAM supports patients in becoming active participants in their own healing and health. Patients may feel more encouraged and focused on improving their QoL when the first phase of treatment is completed [46].

Congruent with other studies [21,47], we found that OD-CAM tends to be superior to SC in reducing patients’ distress and, in particular, patients’ anxiety. Studies have shown a high level of anxiety among patients who seek CAM counseling because they explore every possible treatment option [48]. OD-CAM may thus reduce patients’ levels of anxiety because it includes counseling on both conventional treatment and CAM options. In addition, OD-CAM is an open dialogue, viewing and respecting patients as whole and unique physical, emotional, social and spiritual beings with values, knowledge, preferences and beliefs. Other researchers found that genuinely approaching patients as whole and unique persons leads to emotional well-being [49,50]. Finally, patients in the OD-CAM group may have reported better emotional QoL and lower levels of anxiety because their use of CAM tended to be higher than in the SC group. Use of CAM has in itself been shown to improve emotional well-being [51].

Interestingly, survival tended to be better in the OD-CAM group. Combined with the lack of differences in terms of adverse events, it could be speculated that the supportive care from OD-CAM increases adherence to anti-cancer therapy and thus increases survival. Data collection on the amount of anti-cancer treatment provided during the follow-up period was not part of the study and the results are limited by the short follow-up.

### Strengths and Limitations

The strengths of this trial include the prospective, randomized design with a control group and the systematic data collection using validated patient-reported questionnaires and standardized classifications and registrations of AEs. However, we acknowledge that the study has some important limitations. Due to the phase II design, the study was not powered to fully assess the effect of OD-CAM, and the nature of the intervention did not allow for blinding of patients and data collectors, which may have introduced bias. Moreover, the complexity of the intervention makes it difficult to determine non-specific, specific and mixed effects. However, the improved emotional QoL and increased CAM use within the OD-CAM group can be interpreted as a specific effect of receiving significant counselling about CAM. The potential effects of OD-CAM on adverse events should be interpreted with caution. It is difficult to determine whether adverse events were affected by OD-CAM or actual CAM use. The majority of patients had a positive attitude towards CAM and used CAM prior to enrollment. Although interest in or use of CAM was not an inclusion criterion, the sampling frame might not be representative of patients who are not interested in discussing or using CAM. Fifty-three patients declined to participate in the study due to lack of interest. The potential impact of these patients was therefore, not explored. Furthermore, since most patients had only one session of OD-CAM, the impact of a single versus multiple sessions has not been investigated. The study was conducted in a single cancer center, which might limit the generalizability of the results to other care settings and populations. Furthermore, the rate of decliners may reduce the practicality of the results.

Based on the findings from this study, integration of OD-CAM in daily oncology practice is possible and safe. The findings also suggest that OD-CAM should be conducted when the first phase of antineoplastic treatment is completed, e.g., after two or three cycles of treatment. However, further research on the specific effects of OD-CAM is warranted. The findings of this study will be investigated in a phase III randomized trial (ClinicalTrials.gov: NCT04299451) including qualitative data on OD-CAM experiences to demonstrate how it affects patients and which elements of OD-CAM are important and helpful to them.

## 5. Conclusions

In this study, we have shown that OD-CAM was not superior to SC in reducing the frequency of grade 3–4 AEs but that it did not compromise patient safety. Implementation of OD-CAM may improve the QoL, anxiety and emotional well-being of patients by reducing levels of nausea, vomiting and diarrhea. Finally, OD-CAM provides information about CAM, which potentially improved the patients’ self-care and increased use of CAM. In conclusion, OD-CAM is feasible, safe and clinically important and should be integrated in Danish conventional oncology treatment and care. Research on how OD-CAM potentially contributes to increased conventional oncology treatment adherence and possible improved survival is warranted.

## Figures and Tables

**Figure 1 cancers-14-00952-f001:**
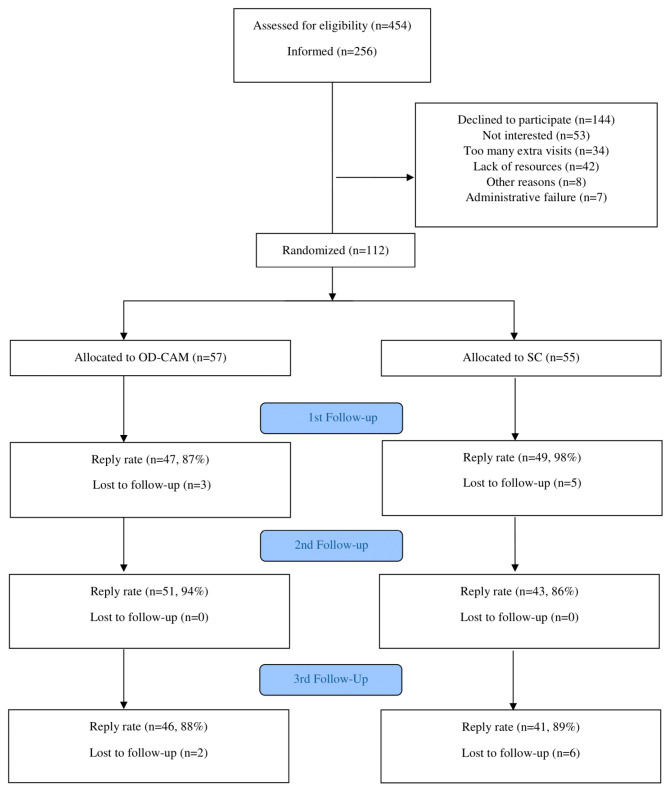
Flow Diagram.

**Figure 2 cancers-14-00952-f002:**
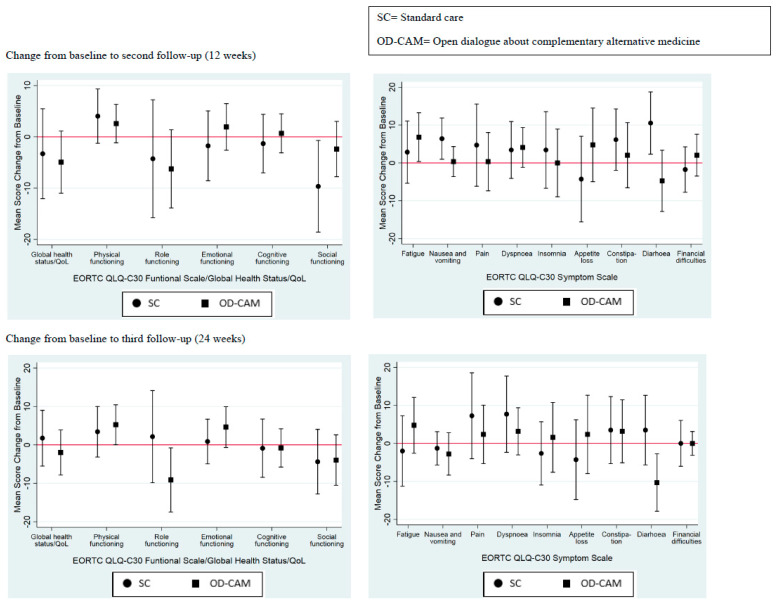
Quality of life.

**Figure 3 cancers-14-00952-f003:**
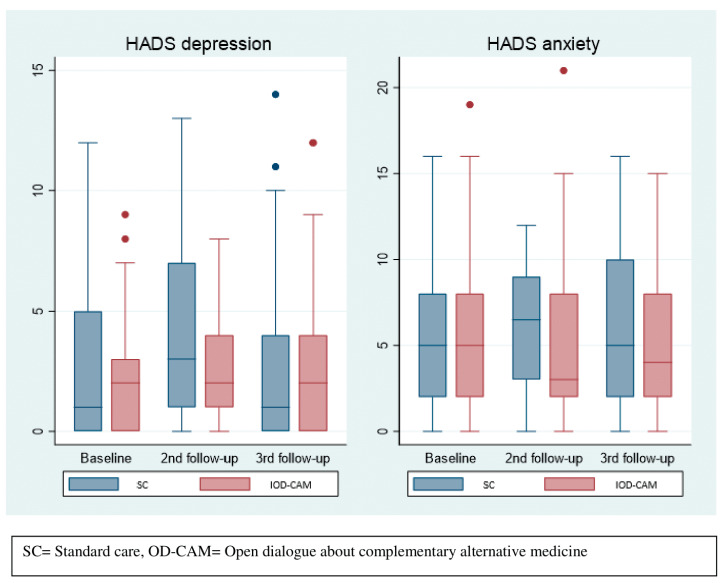
Depression and anxiety.

**Figure 4 cancers-14-00952-f004:**
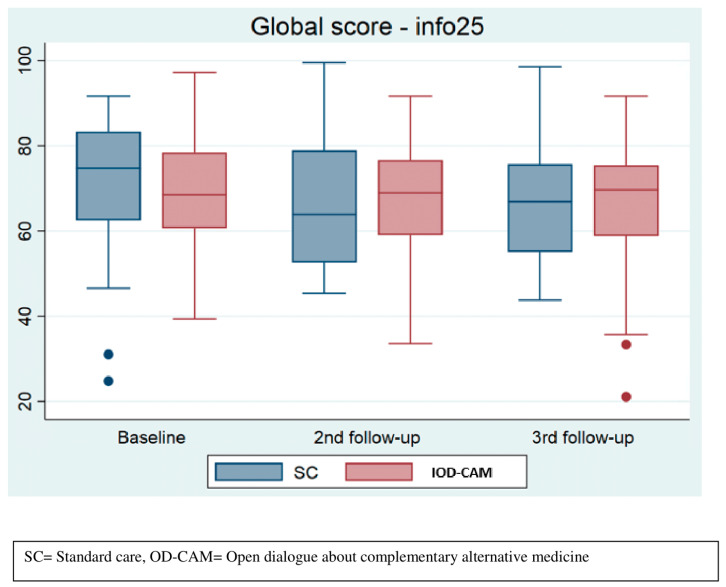
Perceived information.

**Figure 5 cancers-14-00952-f005:**
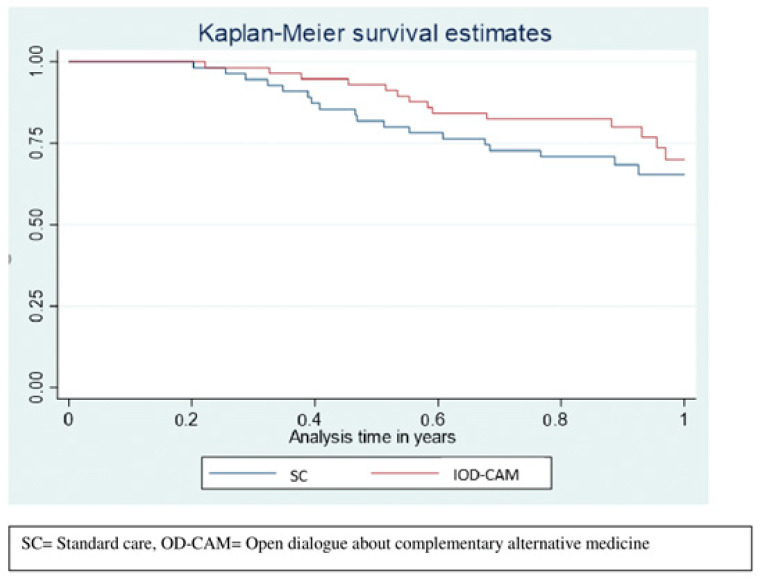
Restricted mean survival.

**Table 1 cancers-14-00952-t001:** Guidelines for open dialogue about complementary alternative medicine: OD-CAM.

**Setting for the OD-CAM**		
**Preparation**	The patient is asked to prepare for the session, including considerations as to current and future use of CAM.	
**Environment**	The OD-CAM takes place in a consultation room designed specifically to provide a healing environment with soft and natural lighting, flowers and relaxing furniture. The room is separate from the clinic.	
**Schedule**	The OD-CAM must be conducted no later than two weeks after randomization and is scheduled to last 60 min.	
**Nurse specialist**	The nurse specialist has completed the program Fellowship in Integrative Medicine at the University of Arizona. This is a training program for health professionals in empowering individuals and communities to optimize health and well-being through evidence-based, sustainable and integrative approaches.	
**Integrative**	Integrative medicine includes a healing-oriented approach, viewing and respecting patients as whole and unique physical, emotional, social and spiritual beings with values, knowledge, preferences and beliefs. It aims to optimize health, quality of life and clinical outcomes and support patients to become active participants in their own healing and health. It emphasizes the therapeutic relationship between health professional and patient. Based on evidence, CAM-information is provided alongside conventional cancer treatment.	
**Content of the OD-CAM**	**In cooperation with the patient**	**Examples of questions to ask**
**1. Understand**	Elicit the patients’ understanding of their situation. Clarify information preferences before asking about CAM use. Ask open questions focusing on psychological/existential issues.	What is your understanding of the situation at this point? What concerns you most about your illness and treatment? What are your hopes for the future?
**2. Respect**	Respect cultural, linguistic and belief diversity. Awareness of attitudes and information needs in relation to models of illness and treatment.	What do you believe might have caused your illness?
**3. Ask**	Ask questions about CAM use. Adopt an inquisitive, open minded and non-judgmental approach. Clarify reasons for asking about CAM.	Are you currently doing or considering doing anything else for your condition/side effects, your overall health or well-being? Are you taking any other medications or treatments? It is very important for me to know about any initiatives you have taken to address your illness so I can help you the best way possible. I am not an expert in this (CAM) but it is important to make sure that any actions or medications you take do not interact negatively with the treatment we give you.
**4. Explore (if the patient is already/considering using CAM)**	Explore the details of CAM use and actively listen. Enquire about current and considered CAM use. Ask about reasons for and expected outcomes of CAM use. Ask about expected outcomes of conventional treatment. Ask if there is a provider of the CAM (if relevant), who it is and what their role will be in relation to the CAM use. Explore the evidence for the CAM’s efficacy and safety. Provide balanced evidential advice in relation to the CAM. Help respond to advice from family and friends (if relevant).	Can you tell me more about this CAM, please? What does it involve? How often do you use it? Have you used it before? What are your reasons for using this CAM? What are you hoping for from this CAM? Has it been helpful so far? How will you know it is helpful for you? Who are you seeing for this CAM? (if relevant) Do you know if there has been any research on the effect of this CAM? Others want the best for you. Let’s talk about these suggestions. What do you think of these suggestions?
**5. Respond**	Respond to the patient’s emotional state, encourage expression of feelings. Express empathy. Support the desire for hope and control; address issues the patient seeks to influence by using CAM (e.g., symptom control, alleviation of side effects, control, desire to live longer).	How are you feeling emotionally? How are you coping with your situation? It sounds like you want to do everything possible. It is natural to feel a need to explore the possible options and I fully support you in that (if relevant).
**6. Discuss**	Discuss relevant concerns about CAM while respecting the patient’s beliefs. Possible concerns: - Caution about substances with unknown effect and quality; - High financial or time cost for CAM with unknown benefits; - Potential for psychological harm. Discuss a reasonable trial period over which an assessment can be made regarding benefits/efficacy of CAM. A symptom diary may help determine whether the CAM is beneficial for the individual patient. Explore alternative ways of addressing the patient’s underlying needs, hopes or fears (especially if there are concerns about the CAM’s potential for harms).	I believe there is little evidence about the benefit or harm associated with this CAM. So, we should be cautious. Might the time involved prevent you from doing other things you like to do? How do you think you might feel if you followed this advice (CAM use) but did not achieve the outcome you hoped for? How long would you expect it to take to see a benefit from this CAM? I can see that you hope this CAM will help you/your cancer/symptoms/side effects/well-being. There are other options we can look at, too. Would you like to hear about them?
**7. Advise**	Encourage use of CAM that may be beneficial. Accept use of CAM for which there is no evidence of physical harm or benefit. Support the decision, even though it conflicts with your private view. Discourage use of CAM where there is no good evidence. It will be unsafe or harmful. Particularly, discourage use of unproven CAM if it is to be used in place of potentially beneficial treatment, especially potentially curative treatment. Balance advice with an acknowledgement of the patient’s rights for self-determination and autonomy.	I recommend this CAM, the evidence suggests that it could help you. We do not know much about this CAM, but it does not seem to be harmful and it may even help you. I respect that is what you wish to do. I have to be honest with you. I am concerned that this CAM may do you greater harm than good. I respect and support your right to make this decision. However, I firmly believe that you have a better chance of a good outcome if you follow this treatment plan. While there is little evidence for us to know if this CAM will be helpful, of course the decision is yours.
**8. Summarize**	Summarize the main points of discussion and check the patient’s understanding. Provide websites and other information or resources, e.g., information about supplements, diet, breathing exercises, yoga, meditation, etc.	We have covered a lot today. Just so that I can check that I have explained things properly, can you summarize what we have discussed? Do you have any further questions or issues you would like to discuss?
**9. Document**	Document the discussion in the patient’s medical record and send a copy to the patient.	I will document what we have discussed today in your medical record and you will receive a copy in your eBoks.
**10. Follow-up**	Follow-up discussion about CAM if relevant.	

**Table 2 cancers-14-00952-t002:** Baseline Sociodemographic and Clinical Characteristics (*n* = 112).

Characteristics	All (*n* = 112)	Intervention Group (*n* = 57)	Control Group (*n* = 55)	*p*
Sex				
Female	72 (64.3%)	32 (56.1%)	40 (72.7%)	0.07 ^a^
Male	40 (35.7%)	25 (43.9%)	15 (27.3%)	
Age (years)				
Mean	62.8	62.3	63.4	0.6
SD	11.23	9.97	12.47	
Married/in a relationship				
Yes	89 (79.5%)	48 (84.2%)	41 (75.5%)	0.21 ^a^
No	23 (20.5%)	9 (15.8%)	14 (25.5%)	
Education				
High	41 (36.6%)	25(43.9%)	16(29%)	0.33 ^b^
Middle	48 (42.9%)	20(35.1%)	28(50.1%)	
Low	14 (12.5%)	7 (12.3%)	7 (12.7%)	
Other	9 (8.0%)	5 (8.8%)	4 (7.3%)	
Diagnosis				
Breast cancer	32 (28.6%)	16 (28.1%)	16 (29.1%)	0.62 ^b^
Prostate cancer	12 (10.7%)	9 (15.8%)	3 (5.5%)	
Lung cancer	32 (28.6%)	15 (26.3%)	17 (30.9%)	
GI cancer	22 (19.6%)	11 (19.3%)	11 (20.0)	
Gynological (uterine + ovarian)	8 (7.1%)	3 (5.3%)	5 (9.1%)	
Pancreas	6 (5.4%)	3 (5.3%)	3 (5.5%)	
Anticancer treatment				
Curative	36 (32.4%)	19 (33.9%)	17 (30.9%)	0.73 ^a,c^
Palliative	75 (67.6%)	37 (66.1%)	38 (69.1%)	
Unknown	1	1	0	
Current treatment				
Chemotherapy (chemotherapy alone, chemotherapy + antibody, chemotherapy + radiation)	100 (89.3%)	51 (89.5%)	49 (89.1%)	0.91 ^b^
Targeted therapy	7 (6.3%)	3 (5.3%)	4 (7.3%)	
(Immunotherapy, antibody)				
Other	5 (4.5%)	3 (5.3%)	2 (3.6%)	
Attitude towards CAM				
Don’t know	2 (1.8%)	1 (1.9%)	1 (1.8%)	0.22 ^b^
Against	3 (2.7%)	2 (3.7%)	1 (1.8%)	
Neutral	19 (17.3%)	9 (16.7%)	10 (17.9%)	
Positive	55 (50%)	22 (40.7%)	33 (58.9%)	
Very positive	31 (28.2%)	20 (37.0%)	11 (19.6%	
Current use of CAM				
Yes	75 (68.2%)	40 (74.1%)	35 (62.5%)	0.19 ^a^
No	35 (31.8%)	14 (25.9%)	21 (37.5%)	
Don’t know	0	0 (0%)	0 (0%)	

^a^ Chi-squared test (binære outcome). ^b^ Fisher’s Exact test. ^c^ Based on patients with known treatment intention (i.e., *n* = 111).

**Table 3 cancers-14-00952-t003:** Frequency of Adverse Events.

		Baseline			First Follow-Up			Second Follow-Up			Third Follow-Up		
		Number (%)		*p*-Value	Number (%)		*p*-Value	Number (%)		*p*-Value	Number (%)		*p*-Value
	Grade	SC	OD-CAM		SC	OD-CAM		SC	OD-CAM		SC	OD-CAM	
**Diarrhea**	0	47 (85.45%)	42 (77.78%)	0.33	35 (74.47%)	37 (82.22%)	0.31	33 (76.74%)	36 (78.26%)	1.00	32 (78.05%)	38 (84.44%)	0.58
	1–2	8 (14.55%)	12 (22.22%)		12 (25.53%)	7 (15.56%)		10 (23.26%)	10 (21.74%)		9 (21.95%)	7 (15.56%)	
	3–4	0 (0.00%)	0 (0.00%)		0 (0.00%)	1 (2.22%)		0 (0.00%)	0 (0.00%)		0 (0.00%)	0 (0.00%)	
**Nausea**	0	31 (56.36%)	34 (61.82%)	0.70	22 (45.83%)	32 (71.11%)	0.02	28 (65.12%)	36 (80.00%)	0.09	30 (73.17%)	33 (73.33%)	1.00
	1–2	24 (43.64%)	21 (38.18%)		25 (52.08%)	13 (28.89%)		15 (34.88%)	8 (17.78%)		11 (26.83%)	11 (24.44%)	
	3–4	0 (0.00%)	0 (0.00%)		1 (2.08%)	0 (0.00%)		0 (0.00%)	1 (2.22%)		0 (0.00%)	1 (2.22%)	
**Peripheral** **motor neuropathy**	0	43 (79.63%)	43 (81.13%)	1.00	37 (78.72%)	32 (71.11%)	0.54	28 (66.67%)	32 (71.11%)	0.82	29 (70.73%)	30 (68.18%)	1.00
	1–2	11 (20.37%)	10 (18.87%)		10 (21.28%)	12 (26.67%)		14 (33.33%)	13 (28.89%)		12 (29.27%)	13 (29.55%)	
	3–4	0 (0.00%)	0 (0.00%)		0 (0.00%)	1 (2.22%)		0 (0.00%)	0 (0.00%)		0 (0.00%)	1 (2.27%)	
**Oral mucositis**	0	45 (81.82%)	43 (79.63%)	0.72	40 (83.33%)	33 (73.33%)	0.27	36 (83.72%)	33 (73.33%)	0.34	36 (87.80%)	36 (80.00%)	0.56
	1–2	9 (16.36%)	11 (20.37%)		8 (16.67%)	10 (22.22%)		7 (16.28%)	10 (22.22%)		5 (12.20%)	8 (17.78%)	
	3–4	1 (1.82%)	0 (0.00%)		0 (0.00%)	2 (4.44%)		0 (0.00%)	2 (4.44%)		0 (0.00%)	1 (2.22%)	
**Mouth** **dryness**	0	33 (60.00%)	29 (52.73%)	0.56	28 (58.33%)	16 (35.56%)	0.03	24 (55.81%)	19 (42.22%)	0.26	27 (65.85%)	25 (55.56%)	0.44
	1–2	22 (40.00%)	25 (45.45%)		20 (41.67%)	27 (60.00%)		19 (44.19%)	24 (53.33%)		14 (34.15%)	19 (42.22%)	
	3–4	0 (0.00%)	1 (1.82%)		0 (0.00%)	2 (4.44%)		0 (0.00%)	2 (4.44%)		0 (0.00%)	1 (2.22%)	
**Constipation**	0	31 (56.36%)	35 (63.64%)	0.56	24 (51.06%)	28 (62.22%)	0.30	22 (51.16%)	30 (65.22%)	0.20	26 (63.41%)	29 (64.44%)	0.82
	1–2	24 (43.64%)	20 (36.36%)		23 (48.94%)	17 (37.78%)		21 (48.84%)	16 (34.78%)		14 (34.15%)	16 (35.56%)	
	3–4	0 (0.00%)	0 (0.00%)		0 (0.00%)	0 (0.00%)		0 (0.00%)	0 (0.00%)		1 (2.44%)	0 (0.00%)	
**Vomiting**	0	45 (81.82%)	45 (81.82%)	1.00	43 (89.58%)	40 (88.89%)	1.00	41 (95.35%)	39 (86.67%)	0.27	37 (92.50%)	39 (86.67%)	0.72
	1–2	10 (18.18%)	10 (18.18%)		5 (10.42%)	5 (11.11%)		2 (4.65%)	6 (13.33%)		3 (7.50%)	5 (11.11%)	
	3–4	0 (0.00%)	0 (0.00%)		0 (0.00%)	0 (0.00%)		0 (0.00%)	0 (0.00%)		0 (0.00%)	1 (2.22%)	
**Peripheral** **Sensory neuropathy**	0	38 (69.09%)	31 (58.49%)	0.32	34 (72.34%)	23 (51.11%)	0.05	21 (48.84%)	24 (53.33%)	0.83	21 (51.22%)	20 (45.45%)	0.91
	1–2	17 (30.91%)	22 (41.51%)		13 (27.66%)	20 (44.44%)		22 (51.16%)	21 (46.67%)		20 (48.78%)	23 (52.27%)	
	3–4	0 (0.00%)	0 (0.00%)		0 (0.00%)	2 (4.44%)		0 (0.00%)	0 (0.00%)		0 (0.00%)	1 (2.27%)	
**Pain**	0	28 (50.91%)	29 (53.70%)	1.00	26 (56.52%)	25 (55.56%)	1.00	27 (62.79%)	28 (62.22%)	0.91	23 (56.10%)	17 (37.78%)	0.27
	1–2	25 (45.45%)	24 (44.44%)		19 (41.30%)	18 (40.00%)		15 (34.88%)	17 (37.78%)		16 (39.02%)	24 (53.33%)	
	3–4	2 (3.64%)	1 (1.85%)		1 (2.17%)	2 (4.44%)		1 (2.33%)	0 (0.00%)		2 (4.88%)	4 (8.89%)	
**Fatigue**	0	12 (22.22%)	12 (22.64%)	1.00	9 (19.15%)	9 (20.00%)	1.00	10 (23.26%)	11 (23.91%)	1.00	12 (29.27%)	12 (26.67%)	0.91
	1–2	42 (77.78%)	41 (77.36%)		38 (80.85%)	36 (80.00%)		33 (76.74%)	35 (76.09%)		28 (68.29%)	32 (71.11%)	
	3–4	0 (0.00%)	0 (0.00%)		0 (0.00%)	0 (0.00%)		0 (0.00%)	0 (0.00%)		1 (2.44%)	1 (2.22%)	
**Discomfort**	0	34 (62.96%)	36 (69.23%)	0.47	28 (59.57%)	28 (62.22%)	0.59	28 (65.12%)	29 (63.04%)	1.00	27 (65.85%)	29 (64.44%)	1.00
	1–2	20 (37.04%)	15 (28.85%)		17 (36.17%)	17 (37.78%)		15 (34.88%)	16 (34.78%)		13 (31.71%)	15 (33.33%)	
	3–4	0 (0.00%)	1 (1.92%)		2 (4.26%)	0 (0.00%)		0 (0.00%)	1 (2.17%)		1 (2.44%)	1 (2.22%)	
**Febrile neutropenia**	No	-	-	-	45 (93.75%)	42 (95.45%)	1.00	41 (95.35%)	44 (97.78%)	0.61	37 (92.50%)	43 (100.00%)	0.11
	Yes	-	-		3 (6.25%)	2 (4.55%)		2 (4.65%)	1 (2.22%)		3 (7.50%)	0 (0.00%)	
**Hospitalization**	No	-	-	-	37 (77.08%)	33 (73.33%)	0.81	40 (93.02%)	37 (82.22%)	0.20	34 (82.93%)	30 (69.77%)	0.20

SC = Standard care; OD-CAM = Open dialogue about complementary alternative medicine.

## Data Availability

Data in relation to this manuscript are available from the corresponding author upon reasonable request.

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
