# Peer review of "Impact of Open Dialogue about Complementary Alternative Medicine—A Phase II Randomized Controlled Trial"

_cancers, 2022, doi:10.3390/cancers14040952_

Round 1

Reviewer 1 Report

The article is interesting and addresses an important issue, being carried out on a significant number of patients, but requires a clearer explanation of the impact of excluding subjects from the study. The impact of the study aand the conclusion could be more clear present.

Author Response

Dear Reviewer,

We are very grateful for your thorough review and thoughtful suggestions that have helped us to improve the manuscript.

Attached are our responses to your comments.

Best regards,

Mette Stie 

Reviewer 2 Report

The present phase II randomized  trial, Stie et al. demonstarte the significancy of open dialogue in the care of cancer patients by complementary alternative medicine.  In general this is a well-designed study. A total of 112 patients were included in the study, with an average reply rate of 89% in the 3rd follow-up. Finally, the authors showed the evidence that OD-CAM does not compromise the patients’ safety, but could improve the patients’ self-care, treatment adherence and survival.

Major comments:

  1. Would it be more suitable if the primary and secondary outcome measures have been collected at the same follow-up period: 8, 12, 24 weeks?
  2. Line 194: The authors stated that the patients in the OD-CAM group participated in 0-4 sessions with the nurse specialist, whereas no data was demonstrated for the SC-group. If the 0-4 sessions refer to the first recruitment session (baseline) and the 3 follow-ups (8, 12, 24 weeks) is not clear and needs to be specified.
  3. Why was it hypothesized that 25% of the patients in the OD-CAM group would have grade 3-4 AEs eight weeks after enrollment, as compared to 50% in the SC group? An explanation in this regard would be helpful.
  4. The presented data indicates that most of the patients had only one session of open dialogue. Has any analysis been made regarding the impact of a single session versus multiple sessions?
  5. The authors demonstrate in the Kaplan-Meier survival curves that the OD-CAM group patients lived 0.064 years later than the SC-group of patients. Could you provide any additional information what is the impact of disease stage over the survival rate or whether the patients received any additional palliative care?
  6. Line 351: the authors assumed that the OD-CAM could potentially contribute to improved survival. That is quite a strong statement, bearing in mind that in line 318 it is clearly mentioned that the proportion of anti-cancer treatment during the follow-up period was out of the scope of this study.

Minor comments:

  1. Line 28: one should use either number or letters as number of patients (fifty-seven or 57, fifty-five or fifty-five) to keep uniformity. For additional clarity, the OD-CAM and SC could be referred to as OD-CAM group and SC group, respectively.
  2. Line 155: one could include a detailed overview of the validated European Organization for Research and Treatment of 155 Cancer Quality of Life Questionnaire, for example in the supplementary materials.
  3. Table 2 states that 74.1% of patients from the intervention group and 62.5% from control group were using OD-CAM during the study. This information is not mentioned anywhere in the materials and methods.
  4. Line 130: It is not quite clear what does the Standard care include. It would be helpful to shortly describe the major differences between both methods as their relationship toward allopathic medicine and complementary medical care. Does the SC spectrum cover the outpatient care or the inpatient care as well?
  5. Line 199: The authors state “During follow-up, the measurement scores over time was similar in the two groups”. Is the follow-up meant to be the clinical investigation during the 0-4 sessions. Notably it is not quite clear what are measurement scores?
  6. Line 231: The authors write “No statistically significant differences were found within the SC group at the third follow-up”. It is not quite clear whether the differences are between the both OD-CAM and SC-groups or between the follow-ups within the SC-group. Please do specify.

Author Response

Dear Reviewers,

We are very grateful for your thorough review and thoughtful suggestions that have helped us to improve the manuscript.

Attached are our responses to your comments.

Best regards,

Mette Stie 

Round 2

Reviewer 2 Report

I would like to congratulate the authors. The paper could be published in present form.